# Current Advances and Future Directions of Pluripotent Stem Cells-Derived Engineered Heart Tissue for Treatment of Cardiovascular Diseases

**DOI:** 10.3390/cells13242098

**Published:** 2024-12-18

**Authors:** Xingyu He, Angela Good, Wael Kalou, Waqas Ahmad, Suchandrima Dutta, Sophie Chen, Charles Noah Lin, Karthickeyan Chella Krishnan, Yanbo Fan, Wei Huang, Jialiang Liang, Yigang Wang

**Affiliations:** 1Department of Pathology and Laboratory Medicine, College of Medicine, University of Cincinnati, Cincinnati, OH 45267, USA; hex6@ucmail.uc.edu (X.H.);; 2Department of Internal Medicine, College of Medicine, University of Cincinnati, Cincinnati, OH 45267, USA; 3Department of Pharmacology and Systems Physiology, College of Medicine, University of Cincinnati, Cincinnati, OH 45267, USA; 4Department of Cancer Biology, College of Medicine, University of Cincinnati, Cincinnati, OH 45267, USA

**Keywords:** engineered heart tissue, human pluripotent stem cells, myocardial infarction, genome editing, personalized medicine, cardiovascular therapy, AI integration, tissue engineering

## Abstract

Cardiovascular diseases resulting from myocardial infarction (MI) remain a leading cause of death worldwide, imposing a substantial burden on global health systems. Current MI treatments, primarily pharmacological and surgical, do not regenerate lost myocardium, leaving patients at high risk for heart failure. Engineered heart tissue (EHT) offers a promising solution for MI and related cardiac conditions by replenishing myocardial loss. However, challenges like immune rejection, inadequate vascularization, limited mechanical strength, and incomplete tissue maturation hinder clinical application. The discovery of human-induced pluripotent stem cells (hiPSCs) has transformed the EHT field, enabling new bioengineering innovations. This review explores recent advancements and future directions in hiPSC-derived EHTs, focusing on innovative materials and fabrication methods like bioprinting and decellularization, and assessing their therapeutic potential through preclinical and clinical studies. Achieving functional integration of EHTs in the heart remains challenging due to the need for synchronized contraction, sufficient vascularization, and mechanical compatibility. Solutions such as genome editing, personalized medicine, and AI technologies offer promising strategies to address these translational barriers. Beyond MI, EHTs also show potential in treating ischemic cardiomyopathy, heart valve engineering, and drug screening, underscoring their promise in cardiovascular regenerative medicine.

## 1. Introduction

Cardiovascular diseases (CVDs) continue to rank as a leading cause of death worldwide, posing a substantial burden on global health and healthcare systems [1,2]. Among these, myocardial infarction (MI), commonly referred to as a heart attack, is particularly alarming due to its sudden onset and life-threatening outcomes [3]. MI occurs when blood flow to a region of the heart is obstructed, typically caused by the accumulation of plaques within the coronary arteries—a condition known as atherosclerosis [4,5]. This blockage results in a progressive loss of cardiomyocytes, creating an infarcted area of the heart and, consequently, diminished cardiac function [6]. Because MI can lead to irreversible heart damage, immediate medical intervention is crucial to limit tissue loss.

Current MI treatment strategies primarily involve pharmacological and surgical interventions, each with considerable limitations. Pharmacological treatments, such as antiplatelet agents (e.g., aspirin, clopidogrel) [7,8], anticoagulants like heparin [9], beta-blockers (e.g., metoprolol, atenolol) [10], and ACE inhibitors or angiotensin receptor blockers [11,12], aim to prevent further clot formation, provide oxygen and nutrition, alleviate cardiac stress, and minimize myocardial damage. However, while effective in managing symptoms, these medications are associated with adverse side effects, including bleeding, bradycardia, hypotension, and hyperkalemia, and crucially, they do not reverse existing tissue necrosis [13,14]. In addition, surgical interventions like percutaneous coronary intervention (PCI) and coronary artery bypass grafting (CABG) are aimed at restoring or bypassing obstructed blocked blood flow but come with associated risks, including stent thrombosis, wound infection, and atrial fibrillation. Furthermore, these approaches are often unsuitable for patients with complex comorbidities [15]. In cases of end-stage heart failure, more advanced treatments like intra-aortic balloon pump (IABP), extracorporeal membrane oxygenation (ECMO), or even heart transplantation may be required. While heart transplantation provides a more definitive solution, it remains limited by donor availability and long-term complications such as graft rejection and vasculopathy [16].

Despite some advances, conventional MI treatments do not prevent ventricular remodeling or congestive heart failure (CHF), as they cannot repair the structural damage caused by MI. Following an MI, scar tissue forms within the heart, leading to compromised contractility and unfavorable alterations in cardiac size, shape, and function [2,17]. While treatments can help improve blood flow, they fail to restore lost myocardial function, leaving patients at an elevated risk of developing CHF—a condition that significantly diminishes quality of life through symptoms like dyspnea, fatigue, and fluid retention [18,19]. Furthermore, these approaches inadequately address ischemia-reperfusion injury, where oxidative stress, calcium overload, and inflammation contribute to increased cell death upon blood flow restoration. As a result, there is an urgent need for novel therapeutic strategies capable of not only restoring blood flow but also regenerating damaged myocardial tissue.

In response to these limitations, regenerative medicine and stem cell therapy have emerged as promising alternatives, aiming to replace damaged cardiomyocytes with new, functional cells [20,21]. Cell sources such as mesenchymal stem cells (MSCs) [22], induced pluripotent stem cells (iPSCs) [23], and cardiac progenitor cells have shown potential in enhancing cardiac function. However, challenges such as cell survival, integration with host tissue, and the risk of arrhythmogenesis still limit their efficacy and safety.

Cell therapy, particularly the use of mesenchymal stem cells (MSCs), has gained attention as a potential treatment for heart disease due to their regenerative properties, such as promoting tissue repair and reducing inflammation [24,25]. However, poor engraftment, restricted differentiation into functional cardiomyocytes, and the potential to promote tumorigenesis are some of the drawbacks of MSC treatment [26]. These difficulties make it less successful as a long-term heart regeneration treatment. By offering a more organized, useful, and repeatable cardiac tissue model, engineered heart tissue (EHT) presents a more promising substitute [27]. EHT integrates cardiomyocytes and other heart-specific cell types [28]. Although MSCs are frequently used as a source of cells or in conjunction with EHT, their paracrine signaling, which is mediated by exosomes or secretomes, is mostly responsible for their therapeutic efficacy. MSCs emit these bioactive chemicals, which have the potential to improve tissue repair, encourage cell survival, and regulate inflammation [29]. However, there are issues with scalability and consistency when using MSCs for exosome or secretome-based treatments. In contrast, EHT can overcome some of the fundamental drawbacks of MSC-based therapies by including these bioactive chemicals in a more regulated setting, providing a safer and more efficient method for cardiac tissue regeneration [30].

EHT presents a particularly promising development in this field as it leverages advanced tissue engineering techniques to generate functional cardiac tissue for myocardial repair or replacement [31,32]. By seeding iPSC-derived cardiomyocytes (iPSC-CMs) onto biomimetic scaffolds—constructed from materials like collagen, fibrin, or biodegradable polymers—researchers can cultivate tissue that resembles native heart muscle [33]. Furthermore, the use of bioreactors to provide mechanical and electrical stimulation in controlled environments has facilitated the maturation and alignment of cardiomyocytes, enhancing their contractile properties [34,35].

EHT offers a unique advantage over traditional MI therapies by addressing a core limitation: the lack of tissue regeneration. Preclinical studies have demonstrated that EHT patches can integrate with host myocardium, improve cardiac function, and reduce scar tissue [36,37]. For instance, rodent models implanted with EHT patches have shown increased ejection fraction and myocardial wall thickness, indicating significant functional improvement [36,37]. Encouragingly, early-stage clinical trials are exploring the safety and feasibility of EHT implantation in humans, as discussed in a later section of this review.

Unlike conventional treatments that mainly manage symptoms, EHT presents a potential pathway to genuine cardiac repair by replacing damaged tissue [31,38]. Although challenges remain—such as immune rejection, tissue vascularization, and scalability—the potential impact of EHT on MI treatment has spurred considerable research interest. This review will examine the current state of EHT development, its therapeutic applications, and the challenges it faces, highlighting its promise as a transformative therapy for MI and other cardiovascular diseases. The convergence of stem cell biology, biomaterials science, and bioengineering offers hope for creating therapies that may one day restore full cardiac function in MI patients.

## 2. The Development of EHT

The development of EHT began in the early 1990s, a period marked by limited understanding of cardiac cell biology and tissue structure [39]. Early models—primarily monolayer cultures—struggled to replicate the three-dimensional interactions essential for proper heart function. A breakthrough occurred in 1997, when researchers created beating three-dimensional heart tissue using embryonic chicken cardiomyocytes embedded in collagen within a supportive matrix, facilitating natural cell-to-cell interactions [40]. This approach, termed the “lattice method”, embedded cardiomyocytes within a hydrogel matrix to form a structured, contractile tissue. Over time, the cells self-organized, mimicking natural myocardial structure and supporting functional tissue alignment [41].

As EHT research progressed, various approaches were developed to improve tissue structure and function. The “ring method”, for instance, involved casting neonatal rat cardiomyocytes into circular molds, mechanically stretching them to form homogenous, contractile rings of heart tissue [42]. Additional techniques to enhance tissue maturity included mechanical stretching, electrical stimulation, and extending culture time [43,44]. Incorporating growth factors, small molecules, and non-cardiomyocyte cells in co-cultures has further refined EHT quality, helping to replicate the intricate cellular environment of the native myocardium [45].

The introduction of human embryonic stem cells (hESCs) transformed cardiac tissue engineering, enabling the development of EHT with enhanced functionality. Building on this advancement, the emergence of hiPSCs allowed for accurate disease modeling and the creation of patient-specific EHT, further advancing personalized applications. Additionally, bioprinting technologies have revolutionized the field by enabling the precise layering of cells and scaffolds, producing complex, multi-layered cardiac structures that closely resemble natural heart tissue [46].

As research continues to advance, the focus remains on optimizing EHT materials and fabrication methods to improve structural integrity and functionality. These developments bring EHTs closer to clinical application, presenting exciting possibilities for treating MI and other forms of heart disease.

### 2.1. EHT Fabrication

The selection of materials for EHT scaffolds is crucial, as these materials provide structural support, facilitate cell growth, and mirror the biomechanical properties of native heart tissue—key components for effective cardiac repair [47,48]. EHT scaffold materials fall broadly into two categories: natural and synthetic materials, each offering unique benefits and challenges.

#### 2.1.1. Cell Sources

A range of cell sources are used to create EHT, which is essential for reproducing the intricate structure and functionality of the heart. Primary cardiomyocytes, typically isolated from human or animal heart tissue, are among the most widely employed cell types due to their ability to contract and mimic the electrical characteristics of native heart cells [49]. However, their limited availability and donor variability restrict their broader application. Pluripotent stem cells (PSCs), such as embryonic stem cells (ESCs) and induced pluripotent stem cells (iPSCs), provide a renewable and patient-specific source of cells for EHT [50]. These cells are generated through reprogramming somatic cells into a pluripotent state by introducing key transcription factors (e.g., Oct4, Sox2, Klf4, c-Myc), which reset the cells to an embryonic-like stage [51]. Stepwise protocols mimicking embryonic heart development are then employed to differentiate PSCs into cardiomyocytes, typically involving mesodermal induction, cardiac progenitor specification, and maturation into functional cardiomyocytes [52]. These methods ensure high yields of contractile and electrophysiologically active cells suitable for EHT applications.

In addition to cardiomyocytes, supporting cells such as cardiac fibroblasts and endothelial cells are integrated into EHT to replicate the extracellular matrix and vascular structure of the heart, ensuring mechanical and electrical integration [53,54]. Together, these various cell types work in synergy to create functional heart tissues that can be used in drug testing, disease modeling, and regenerative medicine.

#### 2.1.2. Natural Materials

Natural polymers are highly valued in cardiac tissue engineering due to their biocompatibility, bioactivity, and degradability. These properties enable natural polymers to support cell growth, aid in tissue integration, and promote functional recovery in EHT applications [55].

Collagen, the most abundant extracellular matrix (ECM) protein in the body, has proven to be an excellent biomaterial for cardiac engineering. Collagen-based scaffolds provide temporary mechanical support to the ischemic heart, improve cell adhesion, and enhance the retention and engraftment of stem cells—all essential factors for effective myocardial repair and regeneration [56]. For example, Zimmermann et al. developed an EHT by combining neonatal cardiomyocytes with a collagen and Matrigel mix, creating a contractile tissue with vascularization potential [57]. Later improvements led to tissue thicknesses of up to 450 µm, significantly enhancing heart function in animal models [58].

Other natural materials like alginate have also shown promise. Alginate’s biocompatibility and gel-like consistency make it suitable for soft tissue engineering. For instance, an alginate-based EHT model was developed with rat cardiomyocytes embedded within an alginate scaffold, which provided a supportive environment for cell alignment and contractility [59]. In vivo studies showed that alginate-EHT constructs improved myocardial function and reduced infarct size in a rat MI model [60].

Gelatin, another widely used natural polymer, supports cell attachment and proliferation. Gelfoam, a gelatin-based scaffold, has been used to seed cardiomyocytes from fetal rat ventricular muscle, creating three-dimensional cardiac grafts that adhered, proliferated, and formed functional tissue [61]. In addition, gelatin-based materials have also shown promise in valve replacement therapies due to their mechanical compatibility and reduced thrombogenicity, making them versatile candidates for EHT and other cardiac applications [62].

In recent years, hyaluronic acid has gained attention for its modification potential and biocompatibility. Injectable hyaluronic acid microrods have been developed to improve heart function and reduce fibrosis. In a preclinical study by Le et al. (2018) [63], animals treated with these microrods showed increased ejection fraction and improved left ventricular wall thickness compared to controls, without requiring additional cell transplantation.

#### 2.1.3. Synthetic Materials

Synthetic polymers, such as polycaprolactone (PCL), polyglycolic acid (PGA), and polylactic acid (PLA), are also widely used in cardiac tissue engineering. Synthetic scaffolds offer tunable mechanical properties, high scalability, and consistent production quality, making them attractive for large-scale applications [64].

Polyglycolic acid (PGA) and polylactic acid (PLA) are biodegradable and biocompatible, qualities that are essential for long-term integration of cardiac grafts. As these polymers degrade, their byproducts are naturally metabolized, reducing potential toxicity [65]. Both PGA and PLA have received FDA approval for biomedical applications, including tissue engineering and drug delivery, underscoring their safety and efficacy [66]. Additionally, polyurethane stands out for its flexibility, which is necessary to accommodate the contractile nature of cardiac tissue, making it a suitable candidate for EHT applications where mechanical strength and durability are required [67].

In one notable study, researchers used bone marrow-derived mesenchymal progenitor cells in a mixture of collagen and Matrigel seeded onto a polylactic acid mesh. This construct, when sutured into the infarcted region of the heart, helped prevent aneurysmal dilation and improved post-MI remodeling [68]. Both synthetic and natural biomaterials bring valuable qualities to EHT development, and ongoing research aims to refine these materials to enhance their effectiveness in cardiac repair.

### 2.2. Fabrication Methods for EHT

With advancements in cardiac tissue engineering, researchers have developed sophisticated fabrication techniques to create EHTs that structurally and functionally resemble native heart tissue (Figure 1). These methods are essential for advancing the field, contributing to the potential of EHTs in repairing or replacing damaged myocardium. Below are some of the primary fabrication techniques.

Electrospinning is a widely used technique that produces ultra-fine fibrous scaffolds mimicking the ECM of the heart, providing a porous framework that promotes cell attachment, proliferation, and cardiac tissue maturation [69,70,71]. This technique utilizes an electric field to pull a polymer solution into thin fibers, which are then collected on a surface to form a scaffold [72].

A notable example of electrospinning in EHT development is the work by Kai et al., who created a scaffold from PCL and gelatin to replicate the fibrous structure of cardiac ECM and promote cardiomyocyte alignment. Their study demonstrated that electrospun fibers could enhance the organization and elongation of cardiomyocytes, improving the contractile function of engineered tissues. Additionally, the PCL–gelatin scaffold exhibited improved electrical conductivity, essential for synchronized contractions, highlighting its potential for functional heart tissue engineering [73].

Decellularization and recellularization involve removing cells from donor heart tissue, leaving behind a natural ECM scaffold that can be repopulated with patient-derived cells. This approach leverages the biochemical cues within the ECM to support cell attachment, proliferation, and differentiation [74]. This technique allows the scaffold to retain its native structure and biochemical properties, creating a supportive environment for host tissue integration.

Guyette et al. expanded the decellularization approach to human hearts, producing acellular cardiac scaffolds that retained ECM structure and coronary vasculature. These scaffolds were then repopulated with cardiomyocytes derived from hiPSCs and cultured over 120 days, forming tissues with sarcomere structures, contractile force, and electrical conductivity. Although the EHT was not yet clinically applicable, the study demonstrated the promise of decellularized ECM for creating patient-specific heart grafts with minimized immune rejection risks [75].

3D Bioprinting has revolutionized tissue engineering by allowing precise layering of cells and scaffolds to construct complex, multi-layered tissue structures that closely resemble native myocardium. This method can incorporate various cell types and materials, producing EHTs that are more physiologically relevant and suited for applications in cardiovascular research and potential therapeutic use [21,76].

A recent study by Esser et al. introduced a novel 3D bioprinting technique to fabricate functional cardiac tissues by printing hiPSC-CMs within a collagen–hyaluronic acid bioink. This approach created centimeter-scale cardiac constructs, including rings and ventricle-shaped models, which exhibited organized sarcomeres and spontaneous, synchronized contractions lasting over 100 days. Furthermore, these bioprinted tissues were responsive to pharmacological stimuli, indicating functional viability and potential for drug screening. Preclinical studies suggest that bioprinted cardiac tissues can enhance heart function post-MI, which is currently being evaluated in clinical trials (ClinicalTrials.gov (accessed on 16 December 2024) Identifier: NCT04396899).

Beyond traditional 3D printing, 4D bioprinting introduces a dynamic approach, with constructs that change shape over time. Zhang et al. developed a thermo-responsive hiPSC-cardiomyocyte-based construct that transitions from spherical shapes to unfolded patches at body temperature, enabling minimally invasive delivery. This 4D construct demonstrated high biocompatibility, supporting cardiomyocyte proliferation and maturation with optimal mechanical properties and shape recovery, enhancing cell retention and promoting myocardial regeneration after heart injury [77].

### 2.3. Meta-Analysis of Fabrication Techniques

A variety of fabrication techniques are employed in EHT development to address the complex demands of replicating functional cardiac tissue. These techniques must achieve a balance between structural integrity, biocompatibility, and functionality while ensuring scalability and reproducibility for potential clinical applications. Each method leverages distinct principles of tissue engineering to recreate the mechanical, electrical, and vascular properties of the myocardium.

Scaffold-based approaches provide essential structural support for cell adhesion and growth, allowing for the creation of 3D constructs with customizable porosity. However, these methods often face challenges related to inconsistent degradation rates and limited vascularization. In contrast, advanced techniques like 3D bioprinting offer precise spatial control of cell placement and the ability to fabricate complex architectures, enabling the production of vascularized constructs. Despite these advantages, technical complexity and the need for specialized equipment remain significant barriers.

Natural matrices derived from decellularized tissues offer unmatched biological relevance and biomechanical compatibility. These matrices retain native extracellular matrix (ECM) properties, facilitating cell attachment and growth. However, donor variability and the risk of incomplete decellularization present challenges in standardizing this technique. Similarly, hydrogels, with their tunable properties and ability to encapsulate cells, are emerging as versatile materials for both injectable therapies and scaffold-based constructs. Yet, rapid degradation and mechanical mismatches with native tissues limit their efficacy.

Electrospinning, a method for producing fibrous scaffolds that mimic the ECM, excels in creating aligned structures that support cell attachment and enhance electrical conductivity. However, limitations in scalability and solvent toxicity require careful consideration during implementation. Spheroid and organoid formation methods capitalize on self-organization principles to create microtissue constructs, offering new avenues for heart development modeling. Nevertheless, nutrient diffusion limits and heterogeneity remain key obstacles.

Mechanical conditioning methods simulate the physiological mechanical environment of the heart, enhancing tissue strength and durability. These approaches help improve the functional properties of EHT, although they are often equipment-intensive and require precise control of stress parameters. Emerging techniques, such as self-assembling peptide scaffolds, promise biocompatibility and nanostructured tissue development with minimal immune response, though they face challenges related to cost and stability.

Table 1 provides a comparative evaluation of these fabrication techniques, summarizing their respective advantages, potential outcomes, and challenges. This meta-analysis highlights the importance of selecting appropriate fabrication strategies based on the specific goals of EHT development, whether for research, preclinical studies, or future clinical applications.

## 3. Therapeutic Applications and Clinical Progress

EHT offers a promising therapeutic avenue for treating MI by addressing the limitations of conventional therapies. While traditional approaches primarily aim to restore blood flow or manage symptoms, EHT has the unique potential to replace necrotic heart tissue and promote genuine cardiac repair. Recent studies have shown that EHT can mimic the mechanical and electrical properties of natural myocardium, allowing it to integrate with host tissue and support functional heart recovery post-MI [40,90]. Below, we discuss the applications and recent clinical progress of EHT in MI treatment.

### 3.1. EHT in MI Treatment

EHT is designed to replace damaged heart tissue by promoting new cell growth and creating an environment that supports synchronized contraction with the existing myocardium. Preclinical studies have demonstrated the efficacy of EHT in improving cardiac function in MI models (Table 2). These studies highlight EHT’s ability to integrate with host tissue, establish vascular connections, and improve cardiac function. This integration is essential, as successful EHT therapy requires both mechanical and electrical coupling with the host heart to support synchronized beating and overall function [21,28].

Rodent models, particularly mice and rats, are widely used in EHT studies to assess functional improvements following myocardial infarction due to their accessibility and well-characterized genetics. Larger animals, such as pigs and non-human primates, serve as crucial translational models due to their physiological similarities to human hearts. These models allow for the evaluation of EHT’s electrical and mechanical integration with host myocardium and provide critical insights into the potential for clinical translation.

For example, studies in rodents and non-human primates have demonstrated that EHT transplantation leads to notable improvements in heart function metrics, such as ejection fraction and reduced fibrosis. Transplanted EHTs have shown the ability to remuscularize infarcted regions, leading to observable functional recovery. Zimmermann et al. reported significant improvements in systolic and diastolic function, while Weinberger et al. observed enhanced left ventricular ejection fraction and successful tissue engraftment in guinea pig models [58,91]. These findings underscore EHT’s potential to support cardiac repair by actively participating in heart muscle function, as opposed to merely reducing symptoms. By reducing scar tissue formation and improving contractility, EHT may provide a pathway to more effective heart repair for MI patients.

### 3.2. Clinical Trials of EHT

The progression of EHT research from preclinical studies to human clinical trials marks a pivotal step forward in cardiovascular medicine. These clinical trials, though in early stages, are assessing EHT’s potential to treat MI and heart failure by focusing on safety, feasibility, and efficacy. Key areas under evaluation include the immune response to transplanted tissues, the ability of EHTs to integrate structurally and functionally with the host myocardium, and their overall impact on cardiac performance. These early trials are crucial in determining whether EHT can evolve into a reliable regenerative treatment for heart disease, especially as initial findings point to promising outcomes. These clinical trials have been summarized and demonstrated in Table 3.

Clinical trials directly evaluating the use of EHT for MI are limited at present. This is due to the complex requirements for addressing acute damage and the translational challenges of applying EHT in an MI context. However, the listed trials involve related conditions, such as ischemic cardiomyopathy and heart failure, which share pathophysiological similarities with MI. Ischemic cardiomyopathy results from reduced blood flow to the heart muscle, which can lead to MI if untreated, while heart failure is often a consequence of late-stage MI. These studies provide insights into the safety, feasibility, and therapeutic potential of EHT, offering a foundation for future MI-specific trials.

In Japan, the jRCT2053190081 Phase I/II trial is testing allogeneic hiPSC-cardiomyocyte patches in patients with ischemic cardiomyopathy. Early results indicate improved left ventricular function without adverse events such as tumorigenesis, and there is preliminary evidence of increased exercise tolerance. The positive outcomes reported in this initial cohort highlight EHT’s potential to functionally integrate with the heart and enhance overall cardiac performance [99].

Another significant study is the NCT04396899 Phase I/II trial, which is exploring hiPSC-cardiomyocyte EHT for treating heart failure. With 53 patients planned, this trial focuses on assessing whether EHT can effectively remuscularize damaged myocardium on a larger scale. The outcomes from this study will provide critical insights into the scalability and therapeutic effectiveness of EHT for broader clinical use.

The ESCORT trial (NCT02057900), a Phase I trial, tested fibrin patches seeded with hESC-derived cardiac progenitors in patients with severe ischemic left ventricular dysfunction. This pioneering study demonstrated that the procedure was feasible and safe, with no tumor formation or arrhythmias detected. Notably, the patients showed improved systolic motion in treated heart segments, indicating functional benefits. While some participants experienced alloimmunization, this study remains a milestone, showcasing one of the first successful integrations of hESC-derived tissue in human hearts [100].

**Table 3 cells-13-02098-t003:** **Summary of clinical trials exploring the therapeutic potential of EHTs in heart failure and ischemic heart disease**.

Trial ID	Study Type	EHT Type	Condition	Patient Numbers	Major Outcomes	Reference
jRCT2053190081	Phase I/II (Japan)	Allogeneic hiPSC-CM patches	Ischemic Cardiomyopathy	Small initial patient cohort	The patient showed improved left ventricular function with no major adverse events; potential exercise tolerance increase; no tumorigenesis detected	[99]
NCT04396899	Phase I/II	hiPSC-CM EHT	Heart Failure	53 patients planned	Focus on remuscularization of damaged myocardium	[101]
NCT02057900	Phase I (ESCORT Trial)	Fibrin patch with hESC-derived cardiac progenitors	Severe Ischemic Left Ventricular Dysfunction	6 patients	The procedure was feasible and safe, with no tumor formation or arrhythmias observed. Three patients developed alloimmunization. Improved systolic motion in the treated segments was noted in most patients. One patient died from heart failure 22 months after surgery.	[100]
NCT05068674	Phase I (Pilot Study)	hESC-CMs	Chronic Left Ventricular Dysfunction	18 patients (planned)	Ongoing study assessing the safety and feasibility of hESC-CMs. Focuses on monitoring adverse cardiac events, arrhythmias, and improvements in heart function (LVEF).	[102]

This table includes trial IDs, study phases, EHT types, patient numbers, and major outcomes. Early findings indicate promising safety profiles, with improvements in heart function and minimal adverse events like tumorigenesis or arrhythmias. Trials aim to evaluate EHT’s efficacy in remuscularizing damaged myocardium and improving cardiac function, supporting its potential as a regenerative treatment in advanced cardiac conditions.

Lastly, the Phase I pilot study NCT05068674 (also known as the HECTOR trial) is currently evaluating the safety and feasibility of hESC-CMs in patients with chronic left ventricular dysfunction. This study, involving 18 patients, focuses on monitoring for adverse cardiac events, arrhythmias, and improvements in left ventricular ejection fraction (LVEF). Early results from this trial are expected to shed light on the long-term functionality of stem cell-derived cardiac tissues in humans, offering valuable data on EHT’s viability in addressing chronic cardiac conditions.

These early clinical trials underscore the potential of EHT to provide functional recovery and integration with host myocardium, with preliminary findings showing enhanced ejection fraction and structural improvements. However, continued research is essential to validate these outcomes, especially to confirm long-term efficacy and safety. As these studies progress, EHT may well become a groundbreaking option for patients with chronic heart failure and post-MI damage, providing a regenerative alternative to traditional cardiac therapies.

## 4. Challenges in EHT Development

While the promise of EHT for treating MI and other cardiovascular conditions is significant, several critical challenges must be addressed to make EHT a viable clinical option. These challenges include immunogenicity, vascularization, scalability, and tissue maturation. Addressing these issues is essential for advancing EHT from research to routine clinical practice.

### 4.1. Immunogenicity and Rejection

One of the primary hurdles in EHT transplantation is the risk of immune rejection. When EHTs are derived from non-autologous cells, the recipient’s immune system may recognize them as foreign and mount an immune response, leading to tissue rejection. This immune response is often triggered by mismatches in human leukocyte antigens (HLAs) between the donor and recipient, which can lead to a robust immune reaction that compromises graft survival [103]. Efforts to address this issue include using patient-specific cells, such as induced pluripotent stem cells (iPSCs), which are derived from the patient’s own tissues and thus less likely to provoke an immune response [104].

In addition to patient-specific approaches, researchers are exploring immunomodulatory biomaterials and gene-editing technologies like CRISPR-Cas9 to reduce the immunogenicity of EHTs. By modifying cell surface markers or creating “universal donor” cell lines, it may be possible to evade immune detection and improve graft compatibility, thereby extending the functional lifespan of EHTs in patients [103].

### 4.2. Vascularization

A crucial challenge in the functional integration of EHTs post-implantation is establishing an adequate blood supply within the tissue. A well-developed vascular network is essential to ensure oxygen and nutrient delivery and waste removal, which are critical for tissue survival and proper function [105]. However, achieving rapid and effective vascularization in engineered tissues remains a significant obstacle, as current methods are still evolving to create interconnected blood vessels within EHTs that can integrate seamlessly with the host’s cardiovascular system [106].

In preclinical studies, several strategies have been explored to enhance vascularization, including co-culturing EHTs with endothelial cells, embedding pro-angiogenic factors within the tissue scaffold, and employing bioprinting techniques to create pre-formed vascular channels. Despite some success, sustaining long-term vascular integration and perfusion in EHTs implanted in larger animal models remains challenging. Ensuring that these engineered tissues can support a robust vascular network is critical to advancing EHT toward effective therapeutic applications.

### 4.3. Scalability

Scaling up EHT production to meet clinical demand is another complex challenge. The process involves not only manufacturing enough tissue but also ensuring that the structural and functional integrity of the tissue is maintained across larger volumes. Achieving scalability presents challenges in bioreactor design, quality control, and supply chain management [28]. Larger-scale production also requires consistent and reproducible methods to ensure that all tissues meet the standards necessary for clinical use.

One potential solution lies in automating aspects of EHT fabrication, such as cell seeding, scaffold assembly, and bioprinting, which could improve reproducibility and reduce labor costs. Moreover, designing advanced bioreactors that can mimic the physiological environment more closely, including appropriate mechanical and electrical stimulation, could enhance tissue maturation on a larger scale. Addressing these scalability challenges is essential for making EHT a feasible therapy for a broader patient population.

### 4.4. Tissue Maturation and Functional Assessments

Another significant challenge in EHT development is achieving tissue maturation comparable to native myocardium. Unlike other tissues, cardiac tissue requires extensive structural and functional maturity, as well as electrophysiological and metabolic stability, before it can perform effectively within the heart [55,107]. EHT maturation typically involves promoting the organization of sarcomeric proteins, enhancing electrical coupling through connexin-43, and fostering metabolic transitions from glycolysis to oxidative phosphorylation [108].

Functional assessments are equally critical for evaluating the readiness of EHTs for clinical application. These evaluations include measuring contractile force, electrophysiological properties, and metabolic activity. Researchers commonly use electrocardiographs (EKG) to monitor electrical activity and assess how closely EHT replicates native heart function, though additional techniques such as calcium imaging and contractility measurements offer deeper insights into tissue performance [109]. However, achieving functional maturity remains a work in progress, as EHTs often lack the biological complexity seen in native myocardium, including a full complement of fibroblasts, smooth muscle cells, and endothelial cells [110,111].

In summary, while EHT represents a groundbreaking approach to cardiac repair, overcoming challenges in immunogenicity, vascularization, scalability, and tissue maturation is essential for translating these innovations from bench to bedside. Continued research focused on these critical aspects will help pave the way for EHT to become a viable therapeutic option for heart disease soon.

### 4.5. Risk of Tumorigenesis and Arrhythmogenesis

There are possible dangers associated with the use of engineered heart tissue (EHT), especially those related to cancer and arrhythmogenesis [42]. When undifferentiated stem cells, which are frequently employed to produce cardiomyocytes for EHT, retain the capacity to multiply unchecked, tumorigenesis may result [112]. These cells may remain in the engineered tissue and aid in the growth of malignancies if they are not completely differentiated or are not appropriately controlled. Similar to this, arrhythmogenesis is a serious issue since it can result in aberrant electrical activity when synthetic heart tissue integrates with host cardiac tissue [113]. Areas of asynchronous electrical conduction may result from variations in the maturation and functional integration of cardiomyocytes derived from EHT, which may lead to arrhythmias [114]. To ensure the safety and effectiveness of these treatments in clinical settings, these dangers emphasize the necessity of strict control over the processes of cell differentiation, selection, and maturation in the formation of EHT, as well as close observation for any indications of aberrant cell behavior.

### 4.6. Regulatory Challenges

There are many regulatory obstacles facing the development of engineered heart tissue (EHT), especially when it comes to following international standards and creating clear approval processes for regulatory agencies like the U.S. Food and Drug Administration (FDA). Clinical translation is made more difficult by the FDA’s current absence of a defined regulatory framework for EHT [115,116]. Few of the current guidelines address the special qualities of biologically created tissues; most of them concentrate on medications and medical devices [117]. Because of this, EHT producers have to follow a complicated process: depending on whether the tissue is classified as a biologic or a device, they frequently apply for clearance through the FDA’s more general categories for tissue-based products, such as the 361 or 351 pathways [118].

## 5. Future Directions in EHT Research

While numerous challenges remain in advancing EHT technology, the field shows considerable promise for innovative therapeutic applications. Key directions that stand to enhance EHT’s clinical viability include genome editing and personalized medicine, AI-driven optimizations, and multi-disciplinary collaboration. These perspectives bring valuable insights and opportunities to further this research field.

### 5.1. Genome Editing and Personalized Medicine

Integrating genome editing technologies like CRISPR-Cas9 presents transformative opportunities in EHT development. By enabling precise modifications, genome editing can create customized tissues with reduced immunogenicity and enhanced functionality. For example, CRISPR has been used to modify hiPSC-CMs to upregulate hypoxia-inducible factor 1-alpha (HIF-1α), a transcription factor that helps cells adapt to low oxygen conditions, thus boosting their survival and function in ischemic environments [119,120].

The move toward personalized EHT holds great potential for individualized treatment approaches. By using a patient’s own cells, personalized EHT could drastically reduce rejection risk and improve integration, with the tissue engineered to match patient-specific factors like immune profile and extent of cardiac damage. Genome editing has also shown potential in addressing genetic conditions such as hypertrophic cardiomyopathy (HCM), which is often associated with MYBPC3 gene mutations. By correcting such mutations in patient-specific hiPSCs, researchers could create EHT tailored to treat genetically driven heart diseases [121,122,123].

### 5.2. Machine Learning and AI Integration

Machine learning (ML) and artificial intelligence (AI) open new frontiers in optimizing tissue engineering processes and predicting clinical outcomes. By analyzing large datasets, AI algorithms can identify trends and refine growth conditions for EHT, supporting reproducibility and scalability in tissue production [124]. AI can also enhance personalized medicine by predicting patient-specific responses to different EHT types, guiding tissue selection and engineering strategies [125].

In cardiac safety assessment, deep learning has shown impressive capabilities. For instance, Serrano et al. used AI to analyze action potentials in hiPSC-CMs, training a convolutional neural network (CNN) to identify patterns linked to drug-induced arrhythmia risks. This AI model outperformed traditional metrics by accurately predicting proarrhythmic effects and detecting how specific cardiomyopathic mutations increased sensitivity to drugs [126,127]. Such applications demonstrate AI’s potential in EHT, where it could monitor tissue health, predict arrhythmic risks, and assess tissue functionality, offering a powerful tool for safe, long-term EHT use in regenerative medicine [128,129].

### 5.3. Multi-Disciplinary Collaboration

The advancement of EHT is significantly propelled by multidisciplinary collaboration, integrating expertise from bioengineering, cardiology, computational science, genetics, pharmacology, and materials science. Such collaborative efforts are crucial for addressing complex challenges, including the optimization of scaffold materials and ensuring the biocompatibility and efficacy of EHT [130].

A pivotal area necessitating this interdisciplinary approach is the incorporation of the secretome into EHT. The secretome, consisting of bioactive molecules secreted by cells, plays a vital role in the therapeutic potential of EHT. Key components include growth factors (e.g., VEGF, TGF-β), cytokines, and extracellular vesicles (e.g., exosomes) that mediate paracrine signaling. These factors enhance angiogenesis, modulate immune responses, and promote cell survival in the context of cardiac repair [131]. Advances in biomaterial design have facilitated the integration of secretome components into EHT scaffolds, thereby improving vascularization and functional recovery in preclinical studies [131].

Future directions for EHT research involve leveraging the secretome’s potential as a standalone or adjunct therapy. Bioengineers and materials scientists are instrumental in designing scaffolds that enable the controlled release of secretome components [80], while pharmacologists play a critical role in assessing their safety and efficacy [130]. However, challenges such as standardizing secretome production and ensuring its stability during storage and application persist, necessitating continued interdisciplinary collaboration to overcome these hurdles [80].

Additionally, the exploration of stem cell-free therapies, which utilize the secretome without the direct application of stem cells, represents a promising avenue in cardiac repair. This approach aims to harness the regenerative potential of the secretome while mitigating challenges associated with stem cell therapies, such as immune rejection and tumorigenicity.

Partnerships with clinicians and bioethicists are also essential for translating EHT from laboratory research to clinical applications. Clinicians provide valuable insights into the safety and efficacy standards required for human trials, while bioethicists guide ethical considerations surrounding patient consent, equitable access, and the responsible application of cutting-edge technologies [130]. These collaborations are vital to ensuring that EHT technology, including secretome-based enhancements, is developed responsibly and reaches patients in a safe and equitable manner.

Therefore, the successful development and clinical implementation of EHT are deeply rooted in multidisciplinary collaboration. By integrating the secretome into EHT and exploring stem cell-free therapies, while addressing associated challenges through concerted interdisciplinary efforts, the field can advance toward more effective and ethically sound cardiac repair solutions.

## 6. EHTs Beyond MI

EHTs hold significant promise beyond MI treatment, offering new possibilities in various areas of cardiovascular research and therapy. One prominent application is in addressing ischemic cardiomyopathy, a condition where the heart muscle weakens due to prolonged reduced blood flow. In a pivotal preclinical study, Weinberger et al. transplanted EHTs derived from iPSCs onto the hearts of guinea pigs with chronic ischemia. Over a four-week period, the transplanted EHTs showed improved cardiac function, enhanced vascularization, and no adverse immune reactions, illustrating EHT’s potential to repair damaged myocardium and restore heart function in ischemic conditions [91].

Beyond myocardial repair, EHTs have advanced into the field of heart valve tissue engineering (HVTE), where they are used to develop bioengineered heart valves that can grow and remodel within the patient. Hoerstrup et al. conducted a groundbreaking study, “Functional living trileaflet heart valves grown in vitro”, in which they developed tissue-engineered heart valves by seeding human vascular cells onto biodegradable scaffolds. Cultured in a dynamic bioreactor that mimicked physiological conditions, these engineered valves demonstrated mechanical properties and functionality comparable to native heart valves, highlighting EHT’s potential in creating viable replacements for diseased heart valves [132,133].

EHTs also offer substantial advantages in drug screening and testing, providing more physiologically relevant human cardiac models than traditional systems. In a study by Mannhardt et al., EHTs generated from iPSC-CMs were used to assess contractile responses to various pharmacological agents. These EHTs responded to drugs in a manner consistent with human cardiac tissue, accurately reflecting dose-dependent effects on contractility. This application underscores EHT’s utility in preclinical drug testing, allowing for better predictions of drug efficacy and cardiotoxicity while reducing reliance on animal models [42,134].

Another significant application of EHTs lies in the development of advanced 3D human cardiac tissue models. Ronaldson-Bouchard et al. showcased this potential in their study, “Advanced maturation of human cardiac tissue grown from pluripotent stem cells”, where they created 3D cardiac tissues from human pluripotent stem cells and subjected them to mechanical and electrical stimulation in a specialized bioreactor. This process led to enhanced maturation of the tissues, which displayed adult-like gene expression profiles, electrophysiological properties, and contractile functions. These mature 3D cardiac models are invaluable for studying heart development, genetic disease mechanisms, and therapeutic responses, as well as for personalized medicine applications where patient-specific tissues can be analyzed [135,136,137].

In summary, EHTs extend far beyond MI treatment, showing potential in the repair of ischemic cardiomyopathy, development of functional heart valves, advancement of drug screening processes, and creation of sophisticated 3D cardiac models. These applications not only deepen our understanding of cardiac physiology and pathology but also pave the way for innovative therapies and personalized approaches in cardiovascular health.

## 7. Conclusions

EHT represents a promising frontier in the treatment of MI and other cardiovascular diseases. The evolution of EHT, from its early beginnings in the 1990s to the sophisticated 3D models and bioprinting techniques of today, underscores the remarkable progress that has been made in this field. The ability of EHT to regenerate lost cardiac tissue, a feat unattainable with traditional treatments, offers a beacon of hope for millions of patients worldwide.

While challenges such as immunogenicity, vascularization, and scalability remain, the promising results from preclinical and early clinical trials provide a strong impetus for continued research and development. The integration of cutting-edge technologies like genome editing, machine learning, and AI further expands the horizons of EHT, paving the way for personalized therapies and optimized tissue engineering processes.

The potential of EHT extends beyond MI, with promising applications in other cardiovascular diseases and in drug discovery and testing. As research progresses, we can anticipate a future where EHT becomes a promising treatment modality, transforming the landscape of cardiovascular medicine and offering a new lease of life to countless individuals affected by heart disease.

## Figures and Tables

**Figure 1 cells-13-02098-f001:**
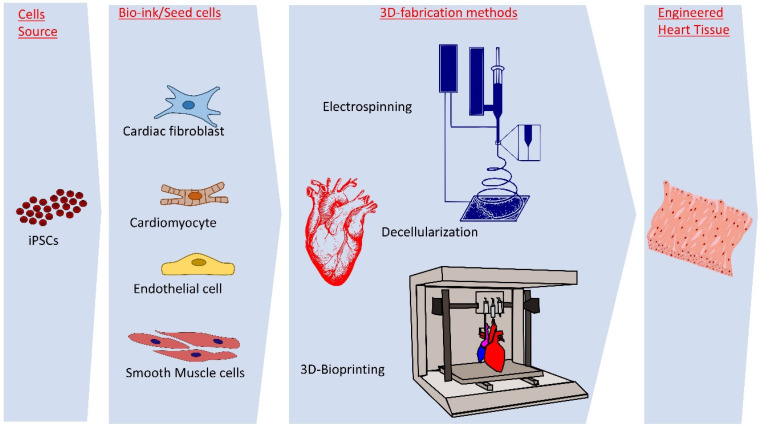
**Schematic representation of cell sources, bio-ink components, fabrication methods, and engineered heart tissue for cardiac tissue engineering**. *The cell sources include iPSCs, which are differentiated into essential cell types such as cardiac fibroblasts, cardiomyocytes, endothelial cells, and smooth muscle cells for bio-ink preparation. Advanced 3D fabrication methods, including electrospinning, decellularization, and 3D bioprinting, are employed to construct engineered heart tissue. These tissues are crucial for applications in regenerative medicine, disease modeling, and cardiac therapy development*.

**Table 1 cells-13-02098-t001:** **Overview of various techniques in EHT development**.

Technique	Cell Types	Advantages	Potential Outcomes	Challenges and Limitations	References
Scaffold-Based Tissue Engineering	CMs; Stem cells; Fibroblasts; Endothelial cells	Provides structural support; Customizable porosity	Formation of 3D constructs; Improved integrity	Immune response; Inconsistent degradation; Limited vascularization	[78,79]
3D Bioprinting	CMs; iPSC-CMs; Endothelial; Smooth muscle cells	Precise cell placement; Complex architectures	Customized tissues; Vascularized constructs	Technical complexity; Specialized equipment; Cell viability	[80,81]
Decellularized Heart Matrices	Stem cells; CMs; Endothelial cells	Natural ECM; Biomechanical compatibility	Tissue regeneration; Organ engineering potential	Donor variability; Incomplete decellularization	[82,83]
Hydrogel-Based Constructs	CMs; Stem cells; Fibroblasts; Endothelial cells	Tunable properties; Supports encapsulation	Injectable therapies; Improved integration	Rapid degradation; Mechanical mismatch	[84]
Electrospinning	CMs; Stem cells	Mimics ECM structure; High surface area	Enhanced alignment; Improved conductivity	Random fiber alignment; Solvent toxicity	[85]
Spheroid and Organoid Formation	CMs; iPSCs; MSCs; Progenitor cells	Enhanced interactions; Self-organization	Microtissue constructs; Heart development modeling	Nutrient diffusion limits; Heterogeneity	[86,87]
Mechanical Conditioning	CMs; Stem cells	Simulates mechanical environment; Enhances strength	Physiological properties; Improved durability	Stress responses; Equipment-intensive	[87,88]
Self-Assembling Peptide Scaffolds	CMs; Stem cells	Biocompatible; Supports adhesion	Nanostructured tissues; Minimal immune response	Costly synthesis; Stability issues	[89]

*Each technique offers unique benefits, such as structural support in scaffold-based engineering, precise cell placement in 3D bioprinting, and biomechanical compatibility with decellularized heart matrices. Potential outcomes include improved tissue integrity, vascularization, and physiological properties. Challenges vary by method, ranging from immune responses and technical complexity to limitations in nutrient diffusion and mechanical mismatch. This table highlights the strengths and limitations of each approach, supporting the strategic selection of methods for specific cardiac regenerative applications*.

**Table 2 cells-13-02098-t002:** **Summary of preclinical studies on EHTs in animal models**.

Study	Animal Model	Cell Source	Experiment Duration	Heart Function Improvement	Fibrosis Reduction	Key Outcomes
Zimmermann et al. (2006) [58]	Rat	Neonatal rat CMs	4 weeks	Significant improvement in LVDP, +dP/dt, and −dP/dt	Not specified	Enhanced systolic and diastolic function
Weinberger et al. (2016) [91]	Guinea Pig	hiPSC-CMs	4 weeks	~7% increase LVEF	Not specified	Successful engraftment; improved cardiac function
Gao et al. (2017) [92]	Mouse	hiPSC-CMs	4 weeks	~10% increase LVEF	Reduced fibrosis area by ~14%	Cell engraftment, wall thickness increased
Riegler et al. (2015) [93]	Rat	hESC-CMs	4 weeks	~10% increase LVEF	Not specified	High engraftment rate, progressive maturation
Kawamura et al. (2012) [94]	Pig	hiPSC-CM sheets	12 weeks	~10% increase LVEF	Not specified	Enhanced heart function; successful cell sheet integration
Chong et al. (2014) [95]	Non-human Primate	hESC-CMs	12 weeks	~16% increase LVEF	Reduced fibrosis area by ~11%	Regeneration of myocardium; manageable arrhythmias; no evidence of human graft ejection
Liu et al. (2018) [96]	Non-human Primates	hESC-CMs	12 weeks	~10% increase LVEF	Reduced fibrosis area by ~10%	Functional recovery; remuscularization observed
Funakoshi et al. (2016) [97]	Mouse	hiPSC-CMs with survival enhancement	12 weeks	~6% increase in FS	Not specified	Improved engraftment; enhanced functional recovery
Sekine et al. (2008) [98]	Rat	Neonatal rat CMs	4 weeks	~10% increase in FS	Not specified	Improved cardiac function; vascularization through cell layering

EHT consistently improved heart function (LVEF, FS) and, in some studies, reduced fibrosis, showing enhanced engraftment, increased myocardial recovery, and functional integration. These findings underscore EHT’s potential for clinical application in cardiac repair.

## Data Availability

No new data were created or analyzed in this study. Data sharing is not applicable to this article.

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
