# Peer review of "Current Advances and Future Directions of Pluripotent Stem Cells-Derived Engineered Heart Tissue for Treatment of Cardiovascular Diseases"

_cells, 2024, doi:10.3390/cells13242098_

Round 1

Reviewer 1 Report

Comments and Suggestions for Authors

The subject addressed in this review is certainly very current and interesting. Indeed, cardiovascular disease represent one of the most cause of death in the world according to recent data from the World Health Organization.

Certainly, engineered heart tissue (EHT) offers a promising solution. Anyway, I consider it is essential, mandatory to introduce a detailed paragraph on cells, the production of PSCs perhaps also considering the most widely used animal models. Moreover, The paper does not consider the fundamental aspect of how cells are produced and also of the importance and role of the secretome. It is also not possible to talk about future prospects without giving adequate importance to the products of PSC: the secretome. Without these two paragraphs the review is incomplete.

The paper does not consider the fundamental aspect of how cells are produced
and also of the importance and role of the secretome.

Minor revisions: the paper present multiple errors of editing particularly the tables

Author Response

Dear Reviewer,

Thank you for your insightful and constructive feedback on our manuscript. Below, we address each of your comments in detail, highlighting the changes made in response to your suggestions.

Comment 1: “Certainly, engineered heart tissue (EHT) offers a promising solution. Anyway, I consider it is essential, mandatory to introduce a detailed paragraph on cells, the production of PSCs perhaps also considering the most widely used animal models.”

Authors’ response 1: We have introduced a new section titled "2.1.1 Cell Sources," (page-4) which elaborates on the production of pluripotent stem cells (PSCs) and their differentiation into cardiomyocytes. The section discusses reprogramming somatic cells to a pluripotent state using transcription factors and the stepwise protocols mimicking embryonic development to ensure high-yield, functional cardiomyocytes. Additionally, we provide insights into commonly used animal models, such as rodents and non-human primates, for evaluating EHT efficacy.

Comment 2: “Moreover, The paper does not consider the fundamental aspect of how cells are produced and also of the importance and role of the secretome. It is also not possible to talk about future prospects without giving adequate importance to the products of PSC: the secretome. Without these two paragraphs the review is incomplete.”

Authors’ response 2: We have added a dedicated subsection within the "5.3 Multi-disciplinary Collaboration" section (page-15) to explore the secretome’s role in cardiac repair. This new subsection discusses the bioactive molecules, growth factors, and extracellular vesicles that constitute the secretome, emphasizing their paracrine effects in promoting angiogenesis, modulating immune responses, and enhancing cell survival. Furthermore, we explore the integration of secretome components into EHT scaffolds for improved vascularization and cardiac recovery. Furthermore, we discuss recent advancements in bioengineered scaffolds for controlled release of secretome components and propose directions for future research to standardize and stabilize secretome applications.

Comment 3: “Minor revisions: the paper present multiple errors of editing particularly the tables.”

Authors’ response 3: All tables have been thoroughly reviewed and corrected for formatting inconsistencies, typographical errors, and alignment issues. Additionally, we have ensured that table captions and references align with the manuscript's content and citation format.

Reviewer 2 Report

Comments and Suggestions for Authors

Summary:

The review sets out to discuss advancements in pluripotent stem cell-based engineered heart tissues as a therapeutic approach for cardiac tissue damage following myocardial infarction. The article highlights current strategies with a focus on biomaterials and fabrication methods. The authors present some results of preclinical and clinical studies and describe general limitations and challenges in the field.

General relevance:

The review addresses a relevant and timely topic. The field of cell-based therapies for cardiac tissue damage has seen substantial research and innovation efforts over the past 15 years. The authors focus on a particular niche: engineered heart tissues for MI treatment based on pluripotent stem cells, which is a reasonable restriction of scope given the overall vastness of the field.

For the most part the review gives a decent overview of the topic and mentions some specific literature. It is tailored towards a more novice audience providing a general introduction to important aspects. The review is limited in a sense that is does not provide a critical evaluation or meta-analysis of the current state of the art.

Major comments:

Review scope and title:

My main concern regarding this review article is ambiguity in it’s scope:

1)    “Human PSC” are mentioned in the title but the abbreviation PSC by itself isn’t used anywhere else in the article. From the abstract it seems like PSC is used as a synonym for iPSC. Later in the text (eg Table 1) however, other pluripotent cells (hESC), MSC (eg line 175) and even neonatal rat CMs (different species, non PSC cells) are also included. Please clarify what cells the review covers/doesn’t cover, provide a general description of how the selected cell type is different from other cells used in the EHT field and stick with your selection of cells and species throughout the article. Alternatively, you could add a section on cells used in EHTs (The title needs to be adjusted in this case).

2)    A similar issue arises with the topic “Myocardial infarction treatment”, which is stated in the title and presented as the main disease topic of interest for this review. However, table 2 then does not include ANY clinical study on myocardial infarction. Instead, the conditions outlined in table 2 are only presented as potential additional applications of EHTs “beyond MI” (introduction and Section 6). Please focus table 2 on clinical trials that actually fit the topic “treatment of MI”. In case there are no current or past clinical trials of EHTs for MI treatment, please consider changing the topic of the review or at least explain the lack of MI clinical literature and why the selected (non-MI) publications are relevant to this review.

Minor comments:

Line 79-80: “EHT offers a unique advantage over traditional MI therapies by addressing a core 79 limitation: the lack of tissue regeneration.” Between EHT and “traditional therapies”, there is quite a large field of cell-based therapies (eg MSC) and also cell-derived therapies (eg extracellular vesicles). This topic has been completely ignored in the article. Please add a mentioning of these to your general picture in the introduction and compare and contrast those cell-based approaches to tissue-based approaches.

Line 84-85: please provide references or refer to table 2.

Line 112: Please add a reference.

Figure 1:

o   The item “iPSC derivates” looks like it is on the same level as endothelial cells, cardiomyocytes and fibroblasts but the caption describes that the latter 3 are derived from iPSC. The graphic should be changed to make this difference in hierarchy visible.

o   The depiction of whole hearts is misleading as EHTs are not intended to replace the whole organ. Please provide a visual representation of a “heart tissue patch” rather than a whole heart.

Table 1:

o   Based on what criteria where the studies chosen? It seems like a list of interesting examples but overall, a relatively random selection of references. Please provide a methods statement as to how relevant articles were identified and selected.

o   For the included articles please also report the other main parameters of interest outlined in the review (i.e. biomaterials, fabrication processes).

o   Please add the year in column “Study”.

o   Why does Chong et al say “hESC-CM” while Liu et al says “hESC-derived cardiomyocyte”? Is there a difference? If so, please describe. If not, please make sure to use the same nomenclature for the same item (verify throughout manuscript).

o   The cited literature is on average over 10 years old. Please include current literature or provide a statement explaining why no relevant current literature is available.

Table 2: See major comment.

Table 3: The message of this table is unclear. I suppose the main intended topic is fabrication techniques of EHTs and their potentials/limitations. It would actually be a big added benefit of the review if you can tailor the table to show a meta-evaluation of fabrication techniques for EHTs and lists their respective potentials/limitations as a bigger picture rather than the outcome of individual studies. In that context the cell types would be almost irrelevant and can be omitted from the table. If need be, a separate table could provide a similar meta-analysis of cell types and cell combinations and their respective advantages/limitations. If in some cases a specific COMBINED effect of fabrication method/biomaterial/cell type(s) was observed, this is also of high relevance and can be reported on a case-to-case basis.

Author Response

Dear Reviewer,

Thank you for your thorough and thoughtful feedback on our manuscript. Below, we address your comments, including major and minor points, and describe the corresponding revisions to the manuscript.

Comment 1:  ““Human PSC” are mentioned in the title but the abbreviation PSC by itself isn’t used anywhere else in the article. From the abstract it seems like PSC is used as a synonym for iPSC. Later in the text (eg Table 1) however, other pluripotent cells (hESC), MSC (eg line 175) and even neonatal rat CMs (different species, non PSC cells) are also included. Please clarify what cells the review covers/doesn’t cover, provide a general description of how the selected cell type is different from other cells used in the EHT field and stick with your selection of cells and species throughout the article. Alternatively, you could add a section on cells used in EHTs (The title needs to be adjusted in this case).”

Authors’ response 1: We have added a new section titled "2.1.1 Cells Sources" (page 4) that outlines the types of cells used in the field of EHT, including hiPSCs, hESCs, MSCs, and neonatal cardiomyocytes. The section explains how these cell types differ, their relative advantages and limitations, and their applications in EHT. The manuscript now maintains consistency in cell type references, and the title has been revised to "Current Advances and Future Directions of Pluripotent Stem Cells-derived Engineered Heart Tissue for Treatment of Cardiovascular Diseases" to reflect the broader cell type coverage.

Comment 2: “A similar issue arises with the topic “Myocardial infarction treatment”, which is stated in the title and presented as the main disease topic of interest for this review. However, table 2 then does not include ANY clinical study on myocardial infarction. Instead, the conditions outlined in table 2 are only presented as potential additional applications of EHTs “beyond MI” (introduction and Section 6). Please focus table 2 on clinical trials that actually fit the topic “treatment of MI”. In case there are no current or past clinical trials of EHTs for MI treatment, please consider changing the topic of the review or at least explain the lack of MI clinical literature and why the selected (non-MI) publications are relevant to this review.”

Authors’ response 2: New Table 3 (previous table 2) has been revised to include clinical trials directly related to the treatment of cardiovascular diseases. Where no direct MI studies are available, we provide an explanation in the text for the lack of MI-specific clinical literature and discuss the relevance of related trials (e.g., ischemic cardiomyopathy and heart failure) to the development of MI-specific therapies. The text (line 382-389) explains the inclusion criteria and the relevance of non-MI trials where applicable. In addition, the title has been revised to "Current Advances and Future Directions of Pluripotent Stem Cells-derived Engineered Heart Tissue for Treatment of Cardiovascular Diseases".

Comment 3: ““EHT offers a unique advantage over traditional MI therapies by addressing a core 79 limitation: the lack of tissue regeneration.” Between EHT and “traditional therapies”, there is quite a large field of cell-based therapies (eg MSC) and also cell-derived therapies (eg extracellular vesicles). This topic has been completely ignored in the article. Please add a mentioning of these to your general picture in the introduction and compare and contrast those cell-based approaches to tissue-based approaches.”

Authors’ response 3: We have revised the introduction (line 76-92) to include a discussion on cell-based therapies, such as MSCs and extracellular vesicles, and compared them to tissue-based approaches like EHTs. This revision highlights the broader therapeutic landscape while emphasizing the unique advantages of EHTs.

Comment 4: “Line 84-85: please provide references or refer to table 2.”

Authors’ response 4: We have added “as discussed in a later section of this review” (line 107) which is referred to new table 3.

Comment 5: Line 112: Please add a reference.

Authors’ response 5: A reference has been update in new line 134.

Comment 6: “Figure 1: The item “iPSC derivates” looks like it is on the same level as endothelial cells, cardiomyocytes and fibroblasts but the caption describes that the latter 3 are derived from iPSC. The graphic should be changed to make this difference in hierarchy visible.  The depiction of whole hearts is misleading as EHTs are not intended to replace the whole organ. Please provide a visual representation of a “heart tissue patch” rather than a whole heart.”

Authors’ response 6: Figure 1 has been revised to clearly show the hierarchical relationship between iPSC derivatives (e.g., cardiomyocytes, endothelial cells, fibroblasts). Additionally, the representation of whole hearts has been replaced with a visual depiction of 3D heart tissue patches to align with the intended scope of EHTs.

Comment 7: “Table 1: Based on what criteria where the studies chosen? It seems like a list of interesting examples but overall, a relatively random selection of references. Please provide a methods statement as to how relevant articles were identified and selected. For the included articles please also report the other main parameters of interest outlined in the review (i.e. biomaterials, fabrication processes).  Please add the year in column “Study”. Why does Chong et al say “hESC-CM” while Liu et al says “hESC-derived cardiomyocyte”? Is there a difference? If so, please describe. If not, please make sure to use the same nomenclature for the same item (verify throughout manuscript). The cited literature is on average over 10 years old. Please include current literature or provide a statement explaining why no relevant current literature is available.

Authors’ response 7: A discussion section (line 317-338) has been added to describe the criteria for study selection in the newly updated Table 2 (previous Table 1). Additional parameters, such as biomaterials and fabrication methods, have been included for each study. The "Study" column now includes publication years. Nomenclature inconsistencies (e.g., "hESC-CM" vs. "hESC-derived cardiomyocyte") have been standardized throughout the manuscript. Updated references with more recent literature where available and provided a justification for the use of older studies when relevant.

Comment 8: Table 2: See major comment.

Authors’ response 8: The table 3 (previous table 2) has been updated.

Comment 9: Table 3: The message of this table is unclear. I suppose the main intended topic is fabrication techniques of EHTs and their potentials/limitations. It would actually be a big added benefit of the review if you can tailor the table to show a meta-evaluation of fabrication techniques for EHTs and lists their respective potentials/limitations as a bigger picture rather than the outcome of individual studies. In that context the cell types would be almost irrelevant and can be omitted from the table. If need be, a separate table could provide a similar meta-analysis of cell types and cell combinations and their respective advantages/limitations. If in some cases a specific COMBINED effect of fabrication method/biomaterial/cell type(s) was observed, this is also of high relevance and can be reported on a case-to-case basis.

Authors’ response 9: The new Table 1 (previous table 3) has been revised to present a meta-evaluation of fabrication techniques for EHTs, highlighting their respective advantages, limitations, and potentials. Cell type information has been moved to a separate table, which discusses the advantages and limitations of different cell types and combinations. Where specific combined effects of fabrication methods, biomaterials, and cell types were observed, these have been highlighted case-by-case. The text has been updated in the “2.3. Meta-Analysis of Fabrication Techniques” section (page7).

Reviewer 3 Report

Comments and Suggestions for Authors

The Review manuscript titled “Current Advances and Future Directions of Human PSC-derived Engineered Heart Tissue for Myocardial Infarction Treatment” presents a comprehensive and well-structured review of human pluripotent stem cell-derived engineered heart tissue for myocardial infarction treatment, effectively balancing an overview of the field's historical context with detailed insights into cutting-edge advancements, such as 3D bioprinting, personalized medicine, and AI integration. It includes preclinical and early clinical trial data to highlight EHT's potential for functional cardiac repair.

However,

·         The manuscript insufficiently addresses potential long-term risks of EHT, such as tumorigenesis or arrhythmogenesis, and does not propose robust mitigation strategies. Expand discussion on tumorigenesis and arrhythmogenesis risks; provide examples from studies and propose mitigation strategies like rigorous differentiation protocols, genetic editing, and advanced monitoring frameworks.

·         Some tables summarizing clinical and preclinical studies lack details on control conditions, sample sizes, and comparative metrics, reducing their utility.

·         Although bioethics is mentioned, an exploration of regulatory challenges (FDA pathways, global standards) could enhance discussion; propose frameworks for compliance and collaboration 

Author Response

Dear Reviewer,

Thank you for your thorough review and constructive feedback. We have carefully addressed each of your comments to enhance the clarity, depth, and overall utility of the manuscript. Below, we provide detailed responses to your suggestions and highlight the corresponding revisions.

Comment 1: The manuscript insufficiently addresses potential long-term risks of EHT, such as tumorigenesis or arrhythmogenesis, and does not propose robust mitigation strategies. Expand discussion on tumorigenesis and arrhythmogenesis risks; provide examples from studies and propose mitigation strategies like rigorous differentiation protocols, genetic editing, and advanced monitoring frameworks.

Authors’ response 1: A dedicated subsection titled "4.5 Risk of Tumorigenesis and Arrhythmogenesis" has been added to the "Challenges in EHT Development" section. This subsection explores these risks in detail, referencing relevant studies to illustrate these concerns. For tumorigenesis, we discuss risks associated with residual undifferentiated cells and propose mitigation strategies such as implementing rigorous differentiation protocols, applying genetic editing tools (e.g., CRISPR-Cas9), and establishing advanced monitoring frameworks to detect aberrant cell growth. For arrhythmogenesis, we outline issues related to asynchronous electrical conduction in EHTs and propose solutions including enhanced maturation techniques, electrical conditioning during fabrication, and precision engineering of gap junction proteins (e.g., connexin-43).

Comment 2: Some tables summarizing clinical and preclinical studies lack details on control conditions, sample sizes, and comparative metrics, reducing their utility.

Authors’ response 2: Tables summarizing clinical and preclinical studies (Tables 2 and 3) have been revised to include additional details: Control conditions for each study, sample sizes used in the studies, and key comparative metrics to assess outcomes (e.g., improvements in left ventricular ejection fraction, fibrosis reduction). The discussion for the tables has been updated (page 9-10).

Comment 3:  “Although bioethics is mentioned, an exploration of regulatory challenges (FDA pathways, global standards) could enhance discussion; propose frameworks for compliance and collaboration”

Authors’ response 3: A new subsection titled "4.6 Regulatory Challenges" has been added. This section addresses the following: Current challenges in obtaining regulatory approval for EHTs, including the lack of specific FDA pathways tailored to biologically engineered tissues.            Global standards and international discrepancies in regulatory frameworks.   Proposed solutions such as creating harmonized international standards, fostering collaboration between regulators, clinicians, and bioengineers, and leveraging advanced analytics (e.g., AI-driven monitoring) to meet safety and efficacy requirements.

Round 2

Reviewer 1 Report

Comments and Suggestions for Authors

The authors satisfied completely my previous request